# Dietary Supplementation with a Blend of Hydrolyzable and Condensed Tannins Ameliorates Diet-Induced Intestinal Inflammation in Zebrafish (*Danio rerio*)

**DOI:** 10.3390/ani13010167

**Published:** 2022-12-31

**Authors:** Roberta Imperatore, Baldassare Fronte, Daniel Scicchitano, Graziella Orso, Maria Marchese, Serena Mero, Rosario Licitra, Elena Coccia, Marco Candela, Marina Paolucci

**Affiliations:** 1Department of Sciences and Technologies, University of Sannio, Via De Sanctis, 2100 Benevento, Italy; 2Department of Veterinary Sciences, University of Pisa, Viale delle Piagge 2, 56124 Pisa, Italy; 3Unit of Microbial Ecology of Health, Department of Pharmacy and Biotechnology, University of Bologna, Via Belmeloro 6, 40126 Bologna, Italy; 4Molecular Medicine & Neurobiology—ZebraLab, IRCCS Foundation Stella Maris, 56128 Pisa, Italy

**Keywords:** zebrafish, intestinal inflammation, cytokines, microbiota, tannins

## Abstract

**Simple Summary:**

The huge amount of fish farmed around the world (about 90 million tons in 2020) requires equally large quantities of feed, which is in a great part of animal origin, as it comes from the capture of aquatic species of little commercial value, such as herring, sardines, and krill. Over the years, this crucial natural resource has been decreasing, calling for alternative sources based on plant products that are cheaper and have fewer fluctuations in price and quantity. However, a plant-based diet causes intestinal inflammation, even in fish that are herbivores, such as carp, one of the most cultivated and consumed cyprinids in the world. Zebrafish is a cyprinid that is widely used as a model for biomedical research and more recently for aquaculture. In this study, it was used to develop intestinal inflammation and evaluate the effects of tannins, polyphenols with antioxidant, anti-inflammatory and immunostimulating properties, in counteracting the intestinal proinflammatory effects of a plant-based diet. The results show that tannins can improve the zebrafish intestinal inflammation caused by a terrestrial-plant-based diet.

**Abstract:**

The current study evaluated the effects of hydrolyzable and condensed tannins from chestnut and quebracho wood, respectively (TSP, Silvafeed^®^), on zebrafish with intestinal inflammation induced by a plant-based diet (basal diet). Four experimental diets were prepared as follows: the basal diet + 0 TSP, the basal diet + TSP at 0.9 g/kg of feed, the basal diet + TSP at 1.7 g/kg of feed, and the basal diet + TSP at 3.4 g/kg of feed. Eighty-four zebrafish (*Danio rerio*) were fed for 12 days with the experimental diets. In zebrafish fed the basal diet, intestine integrity appeared to be altered, with damaged intestinal villi, high immunoexpression of tumor necrosis factor-α (TNFα) and cyclooxygenase 2 (COX2), and high expression of the *cox2*, interleukin 1 *(il-1b*), interleukin 8 (*cxcl8-l1*), and *tnfα* genes. The tannin treatment partially restored intestinal morphology and downregulated the expression of cytokines. The best activity was detected with 1.7 and 3.4 g/kg of feed. In the guts of all groups, Proteobacteria, Fusobacteria, Firmicutes, and Bacteroidetes were the most represented phyla. The most represented genera were *Plesiomonas* and *Sphingomonas*, belonging to the Proteobacteria phylum; *Cetobacterium*, belonging to the Fusobacteria phylum; and *Lactobacillus*, belonging to the Firmicutes phylum. No significant differences were detected among groups, except for a slight decrease in the Fusobacteria phylum and slight increases in the *Shewanella* and *Bacteroides* genera with TSP. In conclusion, these results suggest that tannins can improve the zebrafish intestinal inflammation caused by a terrestrial-plant-based diet in a dose-dependent manner.

## 1. Introduction

In 2020, global production of aquatic animals was estimated to be around 178 million tons, of which aquaculture contributed 49% [1]. The need to produce large quantities of farmed fish has led over the years to a progressive reduction in the aquatic animal resources necessary for the production of fishmeal (FM) [1], prompting scientists to search for alternative protein sources to replace FM. The FM content in the diet can be reduced by using herbal ingredients that are easily available, sustainable, and low-cost. However, the total replacement of FM with plant ingredients induces intestinal inflammation, even in completely herbivorous species such as Cyprinidae carpa gibel (*Carassius auratus gibelio*) [2] and grass carp (*Ctenopharyngodon idella*) [3]. This is a harmful condition that negatively affects feed digestion and nutrient absorption, resulting in impaired fish growth and health [4], leading to economic losses. In fish, the intestinal barrier consists of a single layer of epithelial cells that are selectively permeable to nutrients and secrete chemokines, cytokines, and antimicrobial proteins that are essential for intestinal mucosal immunity, while goblet cells protect the intestinal barriers by producing mucus [5]. At first, the intestinal inflammation can be mild and controllable, but if external stressors, such as feed, persist, the inflammation may turn into more serious epithelial tissue disruption and intestinal dysfunction. As a consequence of the inflammation, epithelial cells produce chemokines to recruit immune cells that in turn produce proinflammatory cytokines such as interleukin (IL-1β) and tumor necrosis factor-alpha (TNFα), leading to the aggravation of inflammation [6]. 

The cyprinidae zebrafish (*Danio rerio*) is a well-established biomedicine and aquaculture research animal model [7,8,9,10]. As a model for aquaculture studies, zebrafish enables lower research costs and possesses a well-known digestive system from both the morphofunctional and microbiome points of view [11,12], making it a useful model to assess the relationship between nutrition and health [13,14]. As an omnivorous species, as reported by field observations of wild zebrafish [15], it can feed on both plant and animal proteins and represents a nutritional animal model for both carnivorous and omnivorous fish [13]. Zebrafish have been used as a fish model for studying the intestinal inflammation induced by plant-based diets [16,17,18], exhibiting the same intestinal inflammatory effects as other fish species. The migration of neutrophils to the affected areas and the production of proinflammatory cytokines take place before the effects are observed histologically (a reduction in villi and an increase in goblet cells) [19,20], shortening study times and giving considerable economic advantage. In a diet-induced intestinal inflammation zebrafish model, the increases in proinflammatory cytokines, neutrophil infiltration, goblet cells, and villi alteration with a loss of mucosal architecture were evident after a 10-day treatment with k-carrageenan [21].

Plant extracts are an economical and sustainable source of bioactive molecules with anti-inflammatory and antioxidant actions, among which polyphenols are the most abundant and widely used in the nutraceutical, cosmetic, and pharmaceutical industries [22]. Tannins are polyphenols, secondary chemicals that are ubiquitous in woody plants. They are classified on the basis of structural characteristics into two main groups: hydrolyzable tannins (HTs) and condensed tannins (CTs), while the third group of tannins, called phlorotannins, is present in brown algae and has less structural complexity than HTs and CTs [23,24,25]. In HTs, phenolic groups such as gallic acid or ellagic acid are linked to a partially or totally esterified carbohydrate, usually represented by D-glucose, giving rise to the esters of the gallic or ellagic acid of glucose. The CTs consist of flavan-3-ol units and oligomeric flavonoids, essentially catechin, epicatechin, gallocatechin, and epigallocatechin bonded via carbon–carbon bonds [23]. Recently, tannins have received considerable attention due to numerous beneficial actions such as antioxidant [26], anticancer [27], antimicrobial, and antiviral [28] activities. However, the potential effects of tannins on human and animal health remain largely unexplored. Research has shown the presence of positive effects linked to the administration of tannins in farmed animals [29]. The most successful supplementation of tannins is attributed to the mitigation of frothy bloat in ruminants [30] and intestinal inflammation in terrestrial farm animals [31]. 

The literature on the effects of tannins on farmed fish production is limited. Previous studies indicated that dietary tannins have a general health-promoting effect in fish, although it may vary in a timing- and dose-dependent manner [32]. Dietary chestnut tannin supplementation had the highest efficacy on growth performance, innate immunity parameters, and antioxidant defenses in juvenile beluga sturgeon (*Huso huso*) [33], Nile tilapia (*Oreochromis niloticus*) [34], common carp (*Cyprinus carpio*) [35], and convict cichlid (*Amatitlania nigrofasciata*) [36] at a concentration of 2 g/kg while diets supplemented with 10, 20, or 30 g/kg of tannic acid resulted in decreases in growth parameters in juvenile European seabass (*Dicentrarchus labrax* L.) [37].

On these bases, the aim of the present study was to explore the effects of commercialized tannin extracts from chestnut (*Castanea sativa*), rich in HT, and quebracho (*Schinopsis spp.*), rich in CT, on zebrafish fed a plant-based diet and bearing intestinal inflammation. The gut histology, the immunoexpression of TNFα and COX2, the cytokine gene expression, and the microbiota composition were analyzed. Altogether, the results suggest that tannins can improve the zebrafish intestinal inflammation caused by a terrestrial-plant-based diet.

## 2. Materials and Methods

### 2.1. Fish Husbandry 

Zebrafish were raised in the “zebrafish facility” of the Department of Veterinary Science of the University of Pisa, Pisa (Italy), and the animal experiments were conducted in agreement with European Union (EU) Directive 2010/63/EU and upon the approval of the Italian Authority for Animal Care and Use Committee (B290E.N.F7X). Eight-month-old wild-type AB-strain zebrafish were used. Fish body weight was measured at the beginning (310,0 ± 118,28 mg; mean ± SD) and end of the trials. Before and during the experimental period (two weeks of acclimation and twelve days of treatments), animals were maintained in a water recirculating system at a temperature of 28 °C, which was monitored daily, as was the pH, electrical conductivity, and dissolved oxygen, as described by Fronte et al. [14], according to the indications of Westerfield [38]. 

### 2.2. Experimental Design and Feeding Protocol

A plant-based diet (control) and three increasing levels of tannins were used. Silvafeed^®^ TSP (Silvateam S.p.A., San Michele Mondovì, Italy), a blend of HTs and CTs obtained from chestnut and quebracho wood, respectively, was included in the plant-based diet. The dietary treatments are shown in Table 1. The ingredients, chemical compositions, and energy contents of the diets were analyzed according to the AOAC (2000) protocol.

The different feeds were prepared as described by Royes and Chapman [39] and Fronte et al. [14]. Briefly, the raw ingredients and TSP were subjected to grinding and homogenization in a mixer, followed by humidification, pelletization, and final drying in a forced-air oven (40 °C for 24 h). At the end, the diets were prepared into convenient pellet sizes (400–600 μm) and stored at 4 °C for further use. Eighty-four zebrafish were randomly divided into 4 dietary treatments (3 replicates each), for a total of 12 tanks (3.5 L capacity; 7 fish per tank). The diets were then supplied to the four dietary groups (*n* = 21 fish per group) for a total of 12 days. The TSP feed supplementation was calculated on the basis of the current literature on cyprinids and other species [33,34,35,36]. It ensured the administration of 0.9, 1.7, and 3.4 g/kg for TSP groups I, II, and III, respectively. For this purpose, the voluntary feed intake (VFI) was measured for 10 days of the two-week adaptation period, when a control diet was supplied ad libitum to all fish four times per day (8:00 A.M., 11:00 A.M., 2:00 P.M., and 5:00 P.M.), following the “five minutes” rule reported by Lawrence [40] (feeding rate: 4.5% of the BW). During the experimental period, the feeds were distributed with the same method. At the end of the experimental period, zebrafish fasted for 24 h and were sacrificed with an overdose of anesthesia (0.25 mg/mL MS-222, Sigma^©^, St. Louis, MO, USA). For each treatment, fish intestines were collected as follows: 36 for the histological and immunohistochemical analysis, 24 for the inflammatory factor analysis, and 24 for the microbiome analysis. 

### 2.3. Histological Analysis

The histological analysis was achieved as described by Orso et al. [21]. Specifically, intestine samples were fixed for 24 h at 4 °C in 4% formalin (pH 7.4) prepared in 0.01 M PBS (phosphate-buffered saline). Dehydration in a series of increasing ethanol grades and clarification with xylene were performed prior to embedding tissues in paraffin. Embedded samples were cut with a microtome (Leica Microsystems, Wetzlar, Germany) into 5 µm sections, and, as reported by Orso et al. [21], anatomically comparable sections of the mid-intestine were selected to be deparaffinized and stained with hematoxylin and eosin (H&E) or Alcian blue to perform morphological analyses and count of the number of goblet cells, respectively. A Leica DMI6000 light microscope equipped with a Leica DFC340 digital camera (Leica Microsystems, Wetzlar, Germany) was used to analyze the stained intestinal sections at 20X and 40X magnifications. The score number was calculated by utilizing the score system proposed by Orso et al. [21], and goblet cell quantification was performed as previously illustrated by Imperatore et al. [32].

### 2.4. Immunohistochemical Analysis

The immunohistochemical technique was performed as described by Imperatore et al. [41]. Briefly, after dewaxing, sections of the mid-intestine were stained with the following primary antibodies: a monoclonal anti-tumor necrosis factor-α (TNFα) antibody raised in mouse (code ab1793, Abcam, Cambridge, UK) or a polyclonal anti-cyclooxygenase 2 (COX2) antibody raised in rabbit (code 69720, NovaTeinBio, Woburn, MA, USA). Briefly, a 5 min incubation with 0.1% H_2_O_2_ was performed to deactivate the endogenous peroxidases and to prepare the sections for the subsequent 30 min incubation with blocking solution (NGS) (10% normal goat serum (Vector Laboratories, CA, USA) and 0.3% Triton X-100 dissolved in 0.1 M Tris-buffered saline, pH 7.6). Subsequently, overnight incubation with primary antibodies prepared in NGS (1:200) was performed at 4 °C. The next day, several washes were performed before the incubation with biotinylated goat anti-mouse or goat anti-rabbit secondary antibodies (Vector Laboratories; 1:100) for 2h at room temperature. In the end, the sections were incubated with the avidin–biotin complex (ABC Kit; Vectastain, Vector Laboratories, CA, USA) for 1h, followed by revelation with 0.05% 3′-diaminobenzidine (DAB) (DAB Sigma Fast, Sigma-Aldrich, St. Louis, MO, USA). The antibody specificity was proven, as reported by Imperatore et al. [41,42]. Specifically, negative controls were performed by omitting the primary antibodies, and they did not show any positivity (Appendix A). A Leica DMI6000 light microscope (Leica Microsystems, Wetzlar, Germany) equipped with a digital camera (JCV FC 340FX, Leica Microsystems, Wetzlar, Germany) was used to acquire digital images under constant light illumination at 20X magnification. 

### 2.5. RNA Isolation, cDNA Synthesis, and Real-Time PCR

The Quick RNA miniprep kit (ZymoResearch, Irvine, CA, USA) was utilized to extract total RNA from zebrafish intestines following the manufacturer’s instructions. By the reverse transcription of about 500 ng of total RNA, using a PrimeScript^TM^ RT Reagent kit (Takara Bio Inc., Shiga, Japan), the cDNA was synthesized, and the qPCRBIO SyGreen Mix Hi-ROX (PCR Biosystem, Wayne, NJ, USA) was employed to perform the quantitative real-time polymerase chain reaction (qRT-PCR), according to Licitra et al. [43]. The sequences of the primers used are listed in Appendix A. The 2^−∆∆Ct^ method was used to calculate the genes’ relative expression levels [44]. The result normalization was performed with respect to the housekeeping gene, β-actin (ENSDARG00000037746). The analysis of gene expression was calculated with the fold-change method. Each assay was performed in triplicate, and 5 samples per group were analyzed. 

### 2.6. Microbiome Analysis

Whole intestine samples from 6 fish for each treatment were used to extract the total DNA that was analyzed, as reported by Pelusio et al. [45]. A NanoDrop ND-1000 (NanoDrop Technologies, Wilmington, DE, USA) was utilized for DNA extraction and quantification. The extracted DNA was stored at –20 °C until further processing [46]. The amplification of the V3-V4 hypervariable regions of the 16S rRNA gene was performed in a 50 uL final volume containing 25 ng of DNA, 2X KAPA HiFi HotStart ReadyMix (Roche, Basel, Switzerland), and 200 nmol/L 341F and 785R primers added with Illumina overhang sequencing adapters. A total of 30 thermal amplification cycles were performed, as described by Pelusio et al. [45]. To purify the PCR products and prepare indexed libraries for Illumina sequencing, the Illumina protocol “16S Metagenomic Sequencing Library Preparation” was followed. In the end, after the normalization to 4 nM, the libraries were pooled, denatured with 0.2 N NaOH, and diluted with a 20% PhiX control to 6 pM. The Illumina MiSeq platform was employed to perform sequencing using a 2 × 250 bp paired-end protocol, as reported in the manufacturer’s instructions (Illumina, San Diego, CA, USA). At the end of the sequencing process, raw sequences were processed using a pipeline combining PANDAseq 2.11 and QIIME2 [47] (https://qiime2.org accessed on 3 March 2017). At the end of the filtering for length (minimum/maximum = 350/550 bp) and quality steps with default parameters using DADA2 [48], high-quality reads were clustered into amplicon sequence variants (ASVs) using the VSEARCH algorithm (2.7.0) [49]. An RDP classifier against the SILVA database was used to define taxonomy [50]. To evaluate intrasample diversity (alpha diversity), we evaluated Faith’s phylogenetic diversity (PD whole tree), the Chao1 index for microbial richness, and the number of observed ASVs. UniFrac distances were used to estimate the beta diversity with a principal coordinates analysis (PCoA). All microbiota analyses and respective plots were produced using R software (https://www.r-project.org/ accessed on 28 November 2020) with the “vegan” (2.5-7) (http://www.cran.r-project.org/package-vegan/ accessed on 28 November 2020) and “Made4” packages (3.14) [51]. A permutation test with pseudo-F ratios (function “Adonis” in “vegan”) was used to test data separation. The Wilcoxon test and the Kruskall–Wallis test were used to evaluate significant differences in alpha diversity and relative taxon abundance between groups, respectively. A *p*-value ≤ 0.05 was deemed to be statistically significant, while *p*-values between 0.05 and 0.1 were considered to be trends.

### 2.7. HPLC Analysis 

An LC-4000 Series Integrated HPLC System (JASCO, Tokyo, Japan) equipped with a liquid chromatography pump (model PU-2829 plus), an autosampler (AS-2059 plus), a column oven (model CO-2060 plus), a UV/Vis Photodiode Array Detector (model MD-2818 plus), and a ChromNAV 2.0 software program (JAsco, Tokyo, Japan) was used to analyze the TSP, as reported by Peng et al. [52]. Samples were loaded onto a C18 Luna column with a 5 μm particle size, 25 cm × 3.00 mm I.D. (Phenomenex, Torrance, CA, USA), with a guard cartridge of the same material. Briefly, the mobile phase was made of water containing 0,2% (*v/v*) phosphoric acid (solvent A) and 82% (*v/v*) acetonitrile containing 0,04% (*v/v*) phosphoric acid (solvent B). The following gradient program was used to run the system: from 0 to 15% B in 15 min, from 15% to 16% B from 15 to 40 min, from 16% to 17% B from 40 to 45 min, from 17% to 43% B from 45 to 48 min, from 43% to 52% B from 48 to 49 min, held isocratic at 52% from 49 to 56 min, reduced from 52% to 43% B from 56 to 57 min, from 43% to 17% B from 57 to 58 min and from 17% to 0% B from 58 to 60 min. The flowrate was 1 mL/min. The injection volume was 20 μL. Peaks were detected at 280 nm and identified by comparison to the retention times of hydrolyzable and condensed tannin standards.

### 2.8. Statistical Analysis

Data related to the fish growth performance and the score number were analyzed using a one-way ANOVA, and differences between groups were tested by a mean HSD Tukey–Kramer test (α = 0.05). The data related to qRT-PCR were first analyzed with the Shapiro–Wilks test to evaluate the normality of the distribution. Post hoc comparisons were performed using a one-way ANOVA. GraphPad Prism 6 (GraphPad Software, Inc., San Diego, CA, USA) was used for all statistical analyses, and *p* ≤ 0.05 was used to define significant differences between treatments.

## 3. Results

### 3.1. Growth Performances

During the experimental period, all fish grew normally, and no statistically significant differences were detected between treatments for the initial and final BWs (Table 2). Similarly, no differences among treatments were observed for BW increment, VFI, or FCR (Appendix A).

### 3.2. Intestinal Histology

Signs of intestinal inflammation were morphologically detectable in zebrafish fed the plant-based diet (control), showing high goblet cell numbers, infiltrated leukocytes, and irregular intestinal villi (Appendix A and Figure 1A). Diets enriched with TSP partially prevented these morphological alterations in a dose-dependent manner. Indeed, intestine sections from zebrafish fed a diet containing TSP at 0.9 g/*k*g of feed (TSP I) showed altered villus morphology, an abundance of goblet cells, and a loss of integrity of the lamina propria (Figure 1B). In zebrafish fed TSP at 1.7 g/*k*g of feed (TSP II) and 3.4 g/kg of feed (TSP III), the integrity of the lamina propria and villi were preserved and reductions in goblet cells and infiltrated leukocytes were observed (Figure 1C,D). Significant reductions in the score number were only found in the TSP II (*p* < 0.05) and TSP III (*p* < 0.0001) groups compared with the control group; while the number of goblet cells only decreased significantly in the TSP III group (*p* < 0.0001) compared with the control group.

### 3.3. Intestinal Immunohistochemistry

Morphological alterations visible in the zebrafish fed a plant-based diet were matched with a high immunoexpression of the proinflammatory marker TNFα in the enteroendocrine and leukocyte infiltrated cells, highlighting the onset of an inflammatory state (Figure 2A). TSP treatment partially reduced TNFα immunoreactivity when added to the diet at high concentrations. Specifically, the intestines of zebrafish fed diets supplemented with TSP at 0.9 g/kg of feed (TSP I) and 1.7 g/kg of feed (TSP II) showed intense TNFα immunoexpression along the villi and in both infiltrates and epithelial cells (Figure 2B,C), demonstrating an intense overt inflammatory state. On the contrary, the intestines of zebrafish fed a diet enriched with TSP at 3.4 g/kg of feed (TSP III) showed a reduction in the immunoexpression of TNFα, which was mainly found in a few infiltrated cells (Figure 2D), indicating an active immune response but the absence of a real inflammatory state. 

COX2 immunoexpression was found mainly confined to the villus epithelium (Figure 3). In particular, the COX2 immunoreactivity was detected on the apical side of epithelial cells in zebrafish fed the plant-based diet (control) (Figure 3A) and the diet containing TSP at 0.9 g/kg of feed (TSP I) (Figure 3B). COX2 expression was reduced in zebrafish fed the diet containing TSP at 1.7 g/kg of feed (TSP II) (Figure 3C) and almost disappeared in zebrafish fed the diet containing TSP at 3.4 g/kg (TSP III) (Figure 3D). 

### 3.4. Inflammatory Factor Analysis

The mRNA expression of *cox2*, *il-1b*, *cxcl8-l1*, and *tnfα* is reported in Figure 4. Zebrafish fed the plant-based diet (control) and the diet containing TSP at 0.9 g/kg of feed (TSP I) showed similar patterns of inflammatory factors, whereas the double and triple dosages of TSP (the 1.7 and 3.4 g/kg of feed of TSP II and III, respectively) significantly reduced the expression of *cox2, il-1b, cxcl8-l1,* and *tnfα* by up to half compared to the control group. 

### 3.5. Intestinal Bacterial Community Profile 

A total of 24 whole intestine samples, yielding 114,329 high-quality reads (mean ± SD, 4764 ± 2719) and clustered into a total of 334 ASVs, were used to perform the 16S rRNA gene sequencing. To assess whether the increasing treatment with TSP could exert a beneficial effect on the gut bacteria community during inflammatory events, the gut microbiome (GM) was analyzed for each dietary group. The variations in the GM profiles (beta diversity) were assessed by a principal coordinate analysis (PCoA) of the unweighted UniFrac distances that were calculated between the samples. In addition, for each dietary group variations in the gut microbial community internal diversity were represented by three different metrics: PD_whole_tree, the Chao1 index, and observed_ASVs. Our findings (Figure 5) showed that none of the TSP groups showed significant variations in the overall GM composition compared to the plant-based-diet group (control) in terms of the whole composition structure (“Adonis”, *p* > 0.05) (Figure 5A–C). However, the TSP III group showed a higher *p*-value (Adonis) compared to the other TSP dietary groups, highlighting that the bacterial community in the TSP III group was more similar to the plant-based-diet group. Conversely, focusing on the microbial internal ecosystem diversity, the diet containing TSP at 3.4 g/kg of feed showed a significant positive effect (Wilcoxon rank-sum test, *p* < 0.05). Indeed, we observed a higher values of internal ecosystem diversity in the TSP II group for all metrics (PD_whole_tree, Chao1, and observed_ASVs) compared to the plant-based-diet group (Figure 5D).

To further assess the GM composition of zebrafish fed different concentrations of TSP, the phylogenetic composition was assessed at the phylum and genus levels, as highlighted in Figure 6A,B, respectively. Overall, similar profiles were found in the GMs of each group in terms of the most abundant bacterial taxa. More specifically, the most abundant phyla were Firmicutes, Fusobacteria, and Proteobacteria, which represented about 94% of the whole intestinal bacterial ecosystem (Figure 6A). On the other hand, *Cetobacterium*, belonging to the Fusobacteria phylum; *Plesiomonas* and *Sphingomonas*, belonging to the Proteobacteria phylum; and *Lactobacillus*, belonging to the Firmicutes phylum were the most represented genera (Figure 6B). 

No statistically significant differences at the genus level (Wilcoxon rank-sum test; *p* > 0.05) were detected between the dietary groups. However, the paired statistical analyses performed between each dietary group showed several tendencies of variations at the genus level. More specifically, the *Bacteroides* genus appeared to be more abundant in both the TSP II and TSP III groups compared to the control group (Wilcoxon rank-sum test; *p* < 0.1). In addition, the *Shewanella* genera appeared to be more abundant in the TSP groups compared to fish fed a plant-based diet (Figure 6B).

### 3.6. HPLC Profile of TSP

The TPS extract was analyzed by HPLC/DAD. The chromatographic profile, reported in Figure 7, shows that TPS is composed of a mixture of HTs and CTs. In particular, by comparing the peak retention times with those of the standards and with the literature, the HTs belong to the ellagitannins, while the CTs correspond to the oligomers and polymers of procyanidins.

## 4. Discussion

In this study, zebrafish were employed as a model to investigate the possible ameliorating effects of a mix of HTs and CTs, the main polyphenols of chestnut and quebracho wood, respectively, on the intestinal inflammation caused by a plant-based diet.

Our results are in agreement with the fish intestinal inflammation caused by a plant-based diet whose detrimental effects on intestinal health status have long been documented in fish. Gut lumen expansion, irregular intestinal villi with the loss of margins, abundant mucus presence, variations in goblet cell numbers and infiltrated leukocytes, the loss of lamina propria integrity, and the accumulation of fat in the submucosa layer are among the most distinctive features of intestinal inflammation [53,54,55] and have been described in several marine and freshwater fish species such as zebrafish [20,56,57], common carp [58], Atlantic salmon (*Salmo salar*) [59,60], gilthead seabream (*Sparus aurata*, L.) [61], silver sillago (*Sillago sihama* Forsskál) [62], and in the hybrid groupers (*Epinephelus fuscoguttatus*×*E. lanceolatu*) [63]. In this study, zebrafish fed a plant-based diet showed altered intestinal morphology and an increased number of goblet cells. Moreover, in agreement with the evidence that the gastrointestinal tract of vertebrates performs important functions associated with immune defense, due to the presence of the gut-associated lymphoid tissue (GALT) [64], leukocyte infiltration was detected. Leukocytes promote the recruitment of other immune cells and help with mucosal recovery by releasing molecules such as cytokines that are crucial for the orchestration of the defense response [65]. 

The addition of tannins to the feed formulation in the presence of intestinal inflammation has been demonstrated to be able to ameliorate the general health status of the intestine by restoring the structure and the organization of the villi almost to normal in terrestrial animal farming [29,66,67]. The tannin amount seems to play a relevant and crucial role in promoting the antioxidant and anti-inflammatory effects. Diet levels of CTs between 10 and 80 g/kg of feed in grass carp (*Ctenopharyngodon idella*) caused impairment of the intestinal immune function and inflammatory status [24], while dietary CTs up to 0.4 g/kg of feed improved the antioxidant status of *L. japonicus* without altering the growth performance [68], and CT up to 1 g/kg of feed mitigated oxidative stress and maintained intestinal health in the spotted sea bass (*Lateolabrax maculatus*) [69]. 

Orso et al. [21] reported the morphological and functional recovery of k-carrageenan-induced intestinal inflammation in zebrafish after treatment with chestnut tannins at 0.3 g/kg of feed, a lower concentration with respect to the present study, where the ameliorative effects of tannins were observed at 1.7 and 3.4 g/kg of feed. This apparent discrepancy could be due to different reasons, one of which is related to the severity of basal inflammation. Based on the classification of inflammation symptoms in zebrafish reported by Orso et al. [21], the intensity of inflammation was more severe in the present study, where the loss of integrity of the villi, the number of goblet cells, and infiltrated leukocytes were quite high in the group fed the plant-based diet. Furthermore, the duration of the proinflammatory stimulus was longer in the present study, in which the specimens were fed the proinflammatory plant-based diet for twelve days, while in the Orso et al. study [21] the inflammatory stimulus was imposed for three days. Another reason is related to the tannins employed in this study, which were a mixture of HTs from chestnut and CTs from quebracho, commercialized under the name of Silvafeed^®^ TSP. An ATR-FTIR analysis of TSP reported by Coccia et al. [70] and the HPLC analysis in the present study confirmed the presence of both CTs and HTs. In particular, according to the literature [71], HTs from chestnut are mainly composed of hydrolyzable ellagitannins, whereas quebracho condensed tannins mainly comprise proanthocyanidin oligomers and polymers consisting of a homologous series of flavan-3-ol-based monomers [72]. However, CTs are not as readily hydrolyzed in the body as HTs and are therefore more difficult to absorb [73]. Thus, the presence of quebracho CTs in this study may explain why the effects were visible at a higher percentage of tannins.

In vivo and in vitro studies in fish have demonstrated that tannins are capable of preventing or ameliorating inflammation by modulating the expression of inflammatory factors [74,75]. Orso et al. [21] reported that chestnut extract, which is rich in tannins, ameliorated K-carrageenan-induced gut inflammation, reducing the expression of *tnfα*, *il-1b*, and *cox2* and increasing the proinflammatory factor *il-10*. Similarly, in the present study, zebrafish fed plant-based diets supplemented with TSP at 1.7 and 3.4 g/kg showed decreases in the proinflammatory factor TNFα and COX2 immunoexpression with respect to the zebrafish fed a plant-based diet, an outcome confirmed by the reduction in the relative expression of the *il-1b*, *cxcl8-l1*, *tnfα*, and *cox2* genes. These results corroborate the beneficial role of tannins in diet-induced inflammation through the reduction in proinflammatory cytokines. Numerous previous studies showed that antinutritional factors contained in plant-based diets, such as vegetal proteins and saponins, induce alterations in growth performance [76], the enhancement of proinflammatory cytokines, and reductions in anti-inflammatory factors in the intestines of both carnivorous fish, such as Atlantic salmon (*Salmo salar* L.) [77], rainbow trout (*Oncorhynchus mykiss*) [78], and orange-spotted grouper (*Epinephelus coioides*) [79], and omnivorous fish, such as Atlantic cod (*Gadus morhua* L.) [80] and zebrafish [17]. Specifically, in line with our results, Perera and Yúfera [17] showed that the inclusion of soybean meal in the zebrafish diet was characterized by high gene expression of *tnf-α*, while Marjara et al. [81] found overexpression of *il-17* and *il-1b* in the intestines of Atlantic salmon (*Salmo salar*) fed a diet containing soybean meal. Moreover, the *cox2* gene expression appeared to be high in Senegalese sole (*Solea senegalensis*) fed a soybean oil diet [82]. 

Tannins, like polyphenols, exert a myriad of effects whose mechanisms of action are not yet fully known. Both CTs and HTs are high-molecular-weight molecules, of which only a small percentage are absorbed at the level of the gastrointestinal system, metabolized by enterocytes, and transferred into the bloodstream, from which they exert systemic effects [83]. The unabsorbed percentage of tannins are used by the intestinal microbiota and degraded into products with low molecular weights that can easily be absorbed [84,85]. Evidence has accumulated on the existence of an interplay between polyphenols and intestinal microbiota. In fact, polyphenols have prebiotic properties that enable them to control the composition and function of the microbiota and antimicrobial properties against pathogenic bacteria, thus demonstrating that they can exert beneficial effects in various disorders of the gastrointestinal system [86]. Metagenomic studies report Proteobacteria, Fusobacteria, Firmicutes, Bacteroidetes, Actinobacteria, and Verrucomicrobia as the principal phyla in the zebrafish gut microbiota and components of the core gut microbiota, as indicated by the comparison of the gut microbiota of wild-caught zebrafish and zebrafish raised in the laboratory [87,88]. Accordingly, the metagenomic analysis conducted in this study showed that the zebrafish intestinal microbiota of all groups were dominated by members of the phyla Proteobacteria, Fusobacteria, and Firmicutes, followed by low percentages of Bacteroidetes and Actinobacteria. 

It has been reported in the literature that in zebrafish with intestinal inflammation Fusobacteria dominate over Proteobacteria, while Firmicutes, Bacteroidetes, and Actinobacteria are less represented [21,89,90]. In this study, Fusobacteria were abundant in zebrafish fed a plant-based diet (which showed signs of intestinal inflammation) and dominated over Proteobacteria. Moreover, in the same group, *Cetobacterium* (belonging to Fusobacteria) was the predominant genera. In humans, Fusobacteria predominate in patients with colorectal cancer and play a pivotal role in promoting the proinflammatory response [91]. On the contrary, Proteobacteria are abundant under normal conditions in the microbiota of fish [92]. An important aspect of this study is that the intestinal inflammation was improved by the presence of tannins, regardless of the microbiota. In fact, none of the tannin-treated groups showed significant variations in the overall microbiota compositions compared to the plant-based-diet group. The treatment with tannins influenced the relative amounts of the phyla, decreasing Fusobacteria and Firmicutes, while increasing Proteobacteria, a result that is in agreement with Orso et al. [21]. Interestingly, the abundance of the Proteobacteria genus *Shewanella* increased in the groups fed TSP at 1.7 and 3.4 g/kg of feed. Some *Shewanella* species can act as fish health modulators thanks to their potential probiotic activity [93]. Thus, a greater presence of this genus after tannin treatment could be considered beneficial for the intestinal gut bacteria community, which could lead to a healthier fish gut microbiota after an inflammatory event. Conversely, *Aeromonas* (Proteobacteria) are pathogenic to aquatic animals and increase in tannin-treated zebrafish. In a study carried out on aquatic bacteria, CTs showed antibacterial activities against *Aeromonas*, an outcome in conflict with this study. However, it is worth noting that CTs had good efficacy to inhibit the growth of pathogenic bacteria, but the efficacy varied with the solvents used to dissolve them, storage and usage temperature, and the acid–base balance [94]. Such variables were not taken into consideration in this study and deserve further attention.

In this study, a decrease in the relative abundance of Firmicutes due to tannins was observed, in agreement with a study carried out in green carp (*Ctenopharyngodon idellus*) treated with HT [95]. However, *Lactobacilli*, beneficial probiotics belonging to Firmicutes, also appeared to decrease with the tannin treatment. A similar result was recently reported by Ke et al. [96] in a study that evaluated the effects of HTs and CTs on the bacterial community of alfalfa silage, reporting that while the application of HTs decreased the abundance of *Lactobacillus,* the opposite results were observed with CTs. The lower absorbability of CTs with respect to HTs [73] may explain the outcome of the present study. 

## 5. Conclusions

Based on the results of the present study, it is possible to conclude that tannins may play a relevant role in counteracting the negative effects of plant-based diet derived inflammation in fish. Ameliorating effects were observed at several levels (gut histology, immunohistochemistry, inflammatory factors, and microbiota). Hence, in the process of improving aquaculture sustainability by reducing the use of fishmeal through replacement with a plant-based meal, the inclusion of tannins in the fish diet may be helpful to maintain fish health. 

## Figures and Tables

**Figure 1 animals-13-00167-f001:**
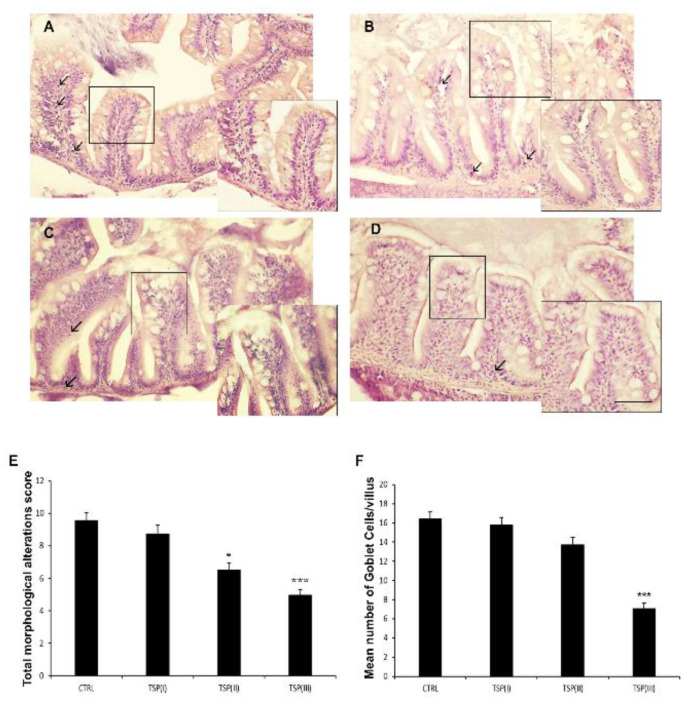
Hematoxylin and eosin (H&E) staining of intestines of (**A**) zebrafish fed a plant-based diet (control); (**B**) TSP I, zebrafish fed a plant-based diet supplemented with TSP at 0.9 g/kg of feed; (**C**) TSP II, zebrafish fed a plant-based diet supplemented with TSP at 1.7 g/kg of feed; and (**D**) TSP III, zebrafish fed a plant-based diet supplemented with TSP at 3.4g/kg of feed. Scale bar: 100 μm; 50 μm for the higher magnifications. Arrows indicate leucocyte infiltrates; linear boxes depict goblet cells. (**E**) Bar graph showing the total morphological alteration score (based on the modification of intestinal folds and gut lumen and increases in the numbers of goblet cells and leukocytes) defined for the intestines of each zebrafish group. Data are expressed as means ± SEs. * *p* < 0.05, *** *p* < 0.0001 compared to the control group. (**F**) Bar graph showing the mean number of goblet cells/villi in the intestines of each zebrafish group. Alcian Blue staining was used to count the goblet cells. Data are expressed as means ± SEs. *** *p* < 0.0001 compared to the control group.

**Figure 2 animals-13-00167-f002:**
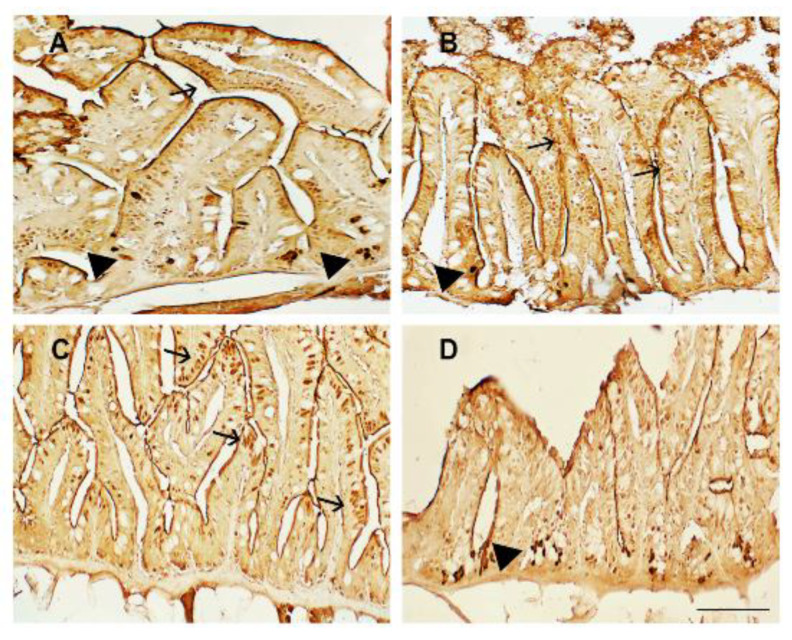
TNFα immunostaining in the intestines of (**A**) zebrafish fed a plant-based diet (control); (**B**) TSP I, zebrafish fed a plant-based diet supplemented with TSP at 0.9 g/kg of feed; (**C**) TSP II, zebrafish fed a plant-based diet supplemented with TSP at 1.7 g/kg of feed; and (**D**) TSP III, zebrafish fed a plant-based diet supplemented with TSP at 3.4 g/kg of feed. Scale bar: 100 μm. Arrows indicate epithelial cells, and black arrowheads indicate infiltrated leukocytes expressing TNFα.

**Figure 3 animals-13-00167-f003:**
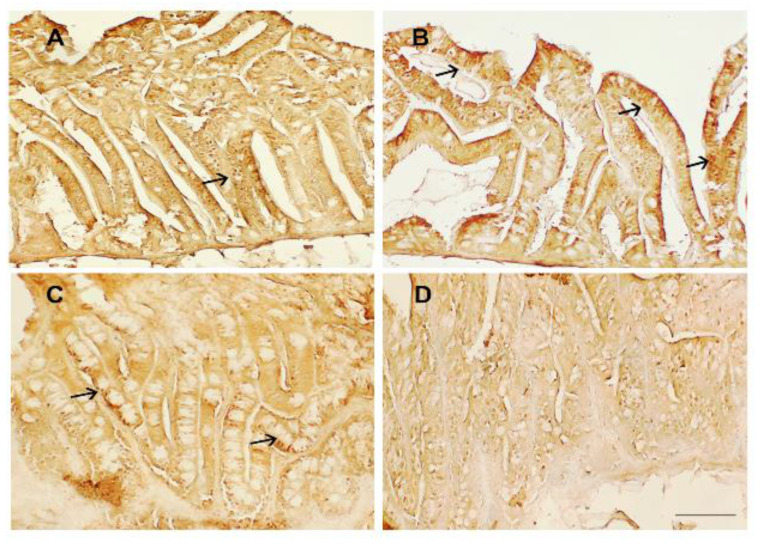
COX2 immunostaining in the intestines of (**A**) zebrafish fed the control diet; (**B**) TSP I, zebrafish fed the control diet supplemented with TSP at 0.9 g/kg of feed; (**C**) TSP II, zebrafish fed the control diet supplemented with TSP at 1.7 g/kg of feed; and (**D**) TSP III, zebrafish fed a diet enriched with TSP at 3.4 g/kg of feed. Scale bar: 100 μm. Arrows indicate COX2 expression on the apical side of epithelial cells.

**Figure 4 animals-13-00167-f004:**
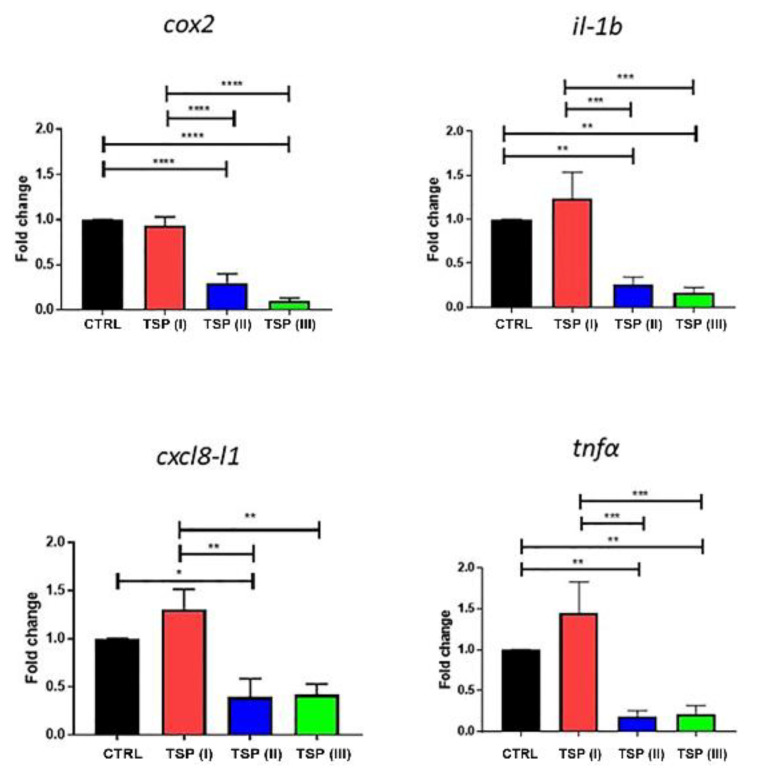
Relative mRNA expression of inflammatory factors in the intestines of zebrafish fed a plant-based diet (control—CTRL) (black column) or a plant-based diet supplemented with TSP at 0.9 g/kg of feed (TSP I), 1.7 g/kg of feed (TSP II), or 3.4 g/kg of feed (TSP III). Only the significant differences are reported. Error bars represent the standard errors of the means (SEMs). **** *p* < 0.00001, *** *p* < 0.0001, ** *p* < 0.001, * *p* < 0.05.

**Figure 5 animals-13-00167-f005:**
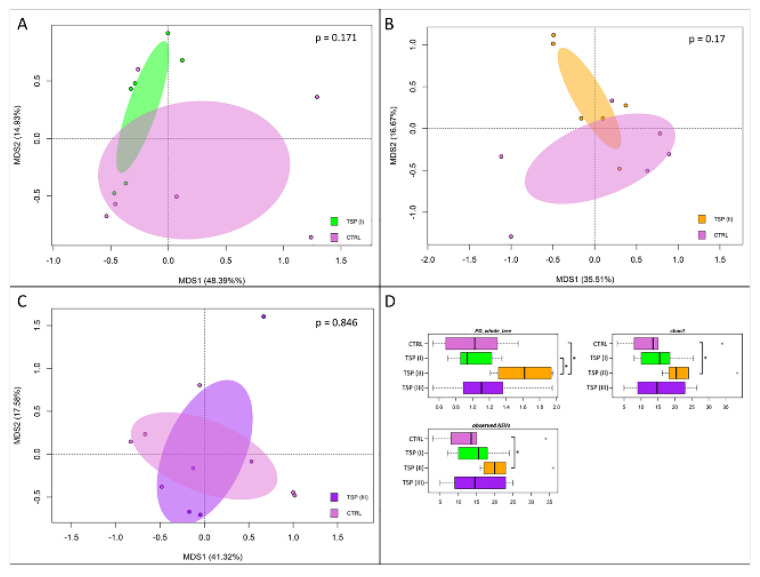
Beta diversity and alpha diversity of the gut microbiota of zebrafish fed a control diet (CTRL), zebrafish fed a diet supplemented with TSP at 0.9 g/kg of feed (TSP I), zebrafish fed a diet supplemented with TSP at 1.7 g/kg of feed (TSP II), and zebrafish fed a diet enriched with TSP at 3.4 g/kg of feed (TSP III). PCoA based on unweighted UniFrac distances between gut microbiota structures of CTRL zebrafish and TSP I (**A**), TSP II (**B**), and TSP III (**C**) zebrafish. In all PCoA plots, samples are not significantly separated (permutation test with pseudo-F ratio, *p* > 0.05). (**D**) Boxplots of alpha diversity, measured with Faith’s phylogenetic diversity (PD_whole_tree), Chao1 index, and observed_ASVs. For only the TSP II group, higher values of alpha diversity were observed for all metrics compared to the plant-based-diet group (Wilcoxon rank-sum test, * *p* ≤ 0.05).

**Figure 6 animals-13-00167-f006:**
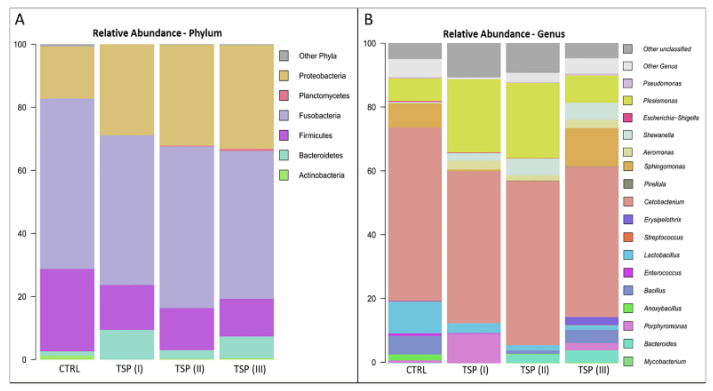
Microbiome compositions of the whole intestines of zebrafish fed a control diet (CTRL), zebrafish fed a diet supplemented with TSP at 0.9 g/kg of feed (TSP I), zebrafish fed a diet supplemented with TSP at 1.7 g/kg of feed (TSP II), and zebrafish fed a diet enriched with TSP at 3.4 g/kg of feed (TSP III). Bar plots summarizing the microbiota compositions at the phylum (**A**) and genus levels (**B**) of CTRL, TSP I, TSP II, and TSP III intestines. Only phyla and genera with relative abundances ≥ 0.5% in at least 2 samples are represented.

**Figure 7 animals-13-00167-f007:**
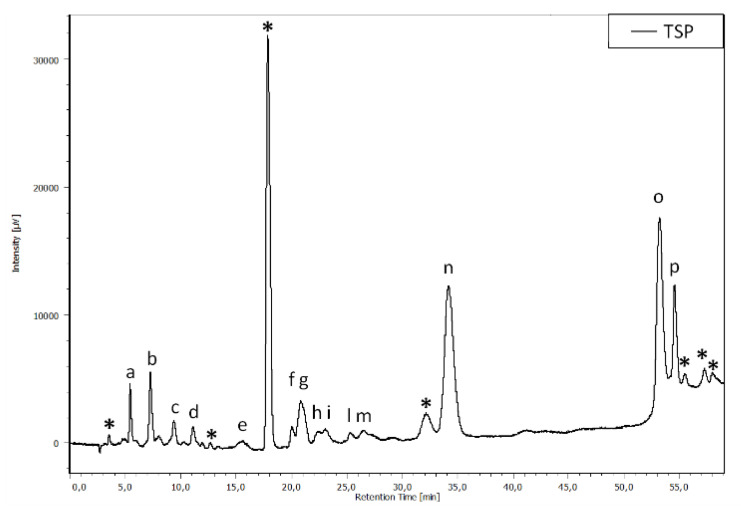
Chromatographic profile, acquired at 280 nm, of the TSP. The asterisks refer to the condensed tannins; the letters (a–p) refer to the hydrolyzable tannins.

**Table 1 animals-13-00167-t001:** Formulations and proximate compositions of the experimental diets (AOAC, 2000).

		Control	TSP I	TSP II	TSP III
Ingredient		%	%	%	%
Soybean meal 48		43.00	43.00	43.00	43.00
Corn meal		22.00	22.00	22.00	22.00
Corn gluten		15.00	15.00	15.00	15.00
Wheat gluten		5.20	5.11	5.03	4.86
Rapeseed meal		5.00	5.00	5.00	5.00
Dicalcium phosphate		3.40	3.40	3.40	3.40
Binder (guar gum)		2.20	2.20	2.20	2.20
Soybean oil		2.00	2.00	2.00	2.00
Choline chloride		1.30	1.30	1.30	1.30
Vitamin and mineral premix ^a^		0.50	0.50	0.50	0.50
Sodium propionate		0.10	0.10	0.10	0.10
L-Lysine		0.10	0.10	0.10	0.10
L-Threonine		0.10	0.10	0.10	0.10
DL-Methionine		0.10	0.10	0.10	0.10
TSP		0.00	0.09	0.17	0.34
Total		100.0	100.0	100.0	100.0
**Proximate Composition**		**As Fed**			
Dry matter	%	96.11	96.1	96.08	96.06
Crude protein	%	36.95	36.88	36.81	36.67
Crude fat	%	4.53	4.53	4.53	4.53
Fiber	%	3.59	3.59	3.59	3.59
Starch	%	18.65	18.64	18.64	18.62
Ash	%	4.99	4.98	4.98	4.98
Gross energy	MJ/kg	17.05	17.03	17.01	16.97

^a^ Vitamin and mineral premix (kg of product): vitamin A = 1,200,000 IU; vitamin D3 = 200,000 IU; vitamin E = 12,000 mg; vitamin K3 = 2400 mg; vitamin B1 = 4800 mg; vitamin B2 = 4800 mg; vitamin B6 = 4000 mg; vitamin B12 = 4800 mg; folic acid = 1200 mg; calcium pantothenate = 12,000 mg; biotin = 48 mg; nicotinic acid = 24,000 mg; Mn = 4000 mg; Zn = 6000 mg; I = 20 mg; Co = 2 mg; Cu = 4 mg; and Se = 20 mg.

**Table 2 animals-13-00167-t002:** Initial and final fish BWs (mg), according to the considered treatments.

Parameters	Initial BW (mg)		Final BW (mg)		BW Increment
Treatments	Mean	SD	Mean	SD	
Control	291.0	116.93	350.1	117.63	20.3%
TSP I	316.9	108.85	375.1	140.84	18.4%
TSP II	307.2	126.57	372.3	120.54	21.2%
TSP III	318.6	121.84	382.9	157.84	20.2%
SEM *	14.617		16.427		
*p*-value	0.9021		0.9031		

* SEM: Standard Error of the Mean.

## Data Availability

The raw data supporting the conclusions of this article will be made available by the authors, without undue reservation.

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
