# Peer review of "Dietary Supplementation with a Blend of Hydrolyzable and Condensed Tannins Ameliorates Diet-Induced Intestinal Inflammation in Zebrafish (Danio rerio)"

_animals, 2022, doi:10.3390/ani13010167_

Round 1

Reviewer 1 Report

The authors of this paper studied the effect of hydrolyzable and condensed tannins on zebrafish with intestinal inflammation caused by a plant-based diet.

The subject is intriguing, and it is currently the focus of research for several fish species.

The introduction is well-organized and provides justification for the experimental design and research questions. I do have some concerns about the level of feeding. The authours estimated a normal level of feeding to be 4.5%, but there is no information about the frequency of meals provided to support this estimate (line 158). The authos provide no results on the level of feeding during the experiments in the following lines. This is an important parameter, and I hope the authors have data on what level was required to reach the ad libitum level in each group.

These are some ideas for improving the text.

-In line 172, the author should specify which part of the intestine was used for all histological analysis (was it the midgut?).

-

Figure 1's Legend should include some information about the parameters included in the score.

-You could use different lines in the in Figure 2. Epithelial and leukocyte arrows

Fig 2 & 3. Could you please clarify that negative controls (no adibodies) were not stained in all the immunohistochemical stained regions of the sections?

Author Response

Dear Reviewer,

Thanks for your comments and suggestions. Here we provide a point-by-point response to your comments.

The authors of this paper studied the effect of hydrolyzable and condensed tannins on zebrafish with intestinal inflammation caused by a plant-based diet.

The subject is intriguing, and it is currently the focus of research for several fish species.

Point 1: The introduction is well-organized and provides justification for the experimental design and research questions. I do have some concerns about the level of feeding. The authours estimated a normal level of feeding to be 4.5%, but there is no information about the frequency of meals provided to support this estimate (line 158). The authos provide no results on the level of feeding during the experiments in the following lines. This is an important parameter, and I hope the authors have data on what level was required to reach the ad libitum level in each group.

Response 1: Thank you for highlighting this point that maybe was not adequately considered during the manuscript elaboration. An estimation of the fish daily feed intake was carried out during the adaptation period, with that aiming to properly assess the TSP inclusion level in the experimental feeds (still to be produced). During the adaptation period, the feed was supplied exactly as described for the experimental period, this meaning 4 time per day (8:00 a.m., 11:00 a.m., 2:00 p.m., and 5:00 p.m.) and according to the mentioned “five minutes rule” described by Lawrence. At the end of the observation period (totally 10 days), the total consumed feed was measured, and the daily feed intake calculated, this resulting to be 4,5% of the fish mean body weight. To better clarify these points, we have modified the related sentences (lines 157-160) and added the table S2 where data related to the BW increment, VFI and FCR observed during the experimental period are shown. Dividing the VFI by the fish Initial BW, it is possible to easily calculate the “feeding rate” that resulted to be 4.5, 4.6, 4.3 and 4.6, respectively, for group Control, TSPI, II and III. Regarding the experimental period, no differences have been observed for BWg, feed intake and FCR and therefore we considered them as “irrelevant” and did not report in the manuscript. By the way, and for avoiding relevant modification to the manuscript, we decided to provide these values as supplementary data (S2).

Point 2: In line 172, the author should specify which part of the intestine was used for all histological analysis (was it the midgut?).

Response 2: Thanks for the suggestion. We have specified in the paragraphs 2.3 and 2.4 which part of the intestine was used for histological and immunohistochemical analysis.

Point 3: Figure 1's Legend should include some information about the parameters included in the score.

Response 3: Thank you for highlighting this point that can help to better understand the differences in the score number. To clarify we added in the figure legend the morphological alterations used to define the score number.

Point 4: You could use different lines in the in Figure 2. Epithelial and leukocyte arrows

Rsponce 4: We changed the figure using arrows for Epithelial cells and arrowhead for leukocytes.

Point 5: Fig 2 & 3. Could you please clarify that negative controls (no adibodies) were not stained in all the immunohistochemical stained regions of the sections?

Response 5: We added a specific sentence in the main text of materials and methods in the paragraph 2.4. Moreover, we added a supplementary figure (Fig S1) showing the negative controls for TNFα and COX2.

Reviewer 2 Report

The study evaluated the effect of hydrolyzable and condensed tannins from chestnut and quebracho wood, respectively, on zebrafish with intestinal inflammation induced by a plant-based diet (basal diet). This study provides a technical reference for HT and CT to improve the intestinal inflammation of cultured fish caused by terrestrial plant diet, which is of great significance.

The following issues need further discussion

1. In Table 1, why is the reduction of wheat gluten consumption considered in the design? Does it affect the intestinal inflammation of fish? Suggestions need to be analyzed and explained in the discussion.

2. On page 7, I suggest that the organizational forms of A and C in Figure 1 should be the same as those of B and D, so that the comparison will be clearer.

3. In "4 Discussion" on page 12, the second paragraph (lines 402-410) is suggested to be introduced in the foreword, which directly discusses the research results.

4. On page 14, lines 493-506, it is unclear whether the content of Fusobacterium and other bacteria that can cause intestinal inflammation in the intestine of zebrafish fed with tannin diet increases or decreases, and whether they have anti-inflammatory or inflammatory effects.

5. It is recommended that only the particularly relevant literature be cited, and some are repeated, such as literature [1] and [2]. Only [1] can be cited.

Author Response

Dear Reviewer,

thanks for your comments and suggestions. Here we provide a point-by-point response to your comments.

The study evaluated the effect of hydrolyzable and condensed tannins from chestnut and quebracho wood, respectively,on zebrafish with intestinal inflammation induced by a plant-based diet (basal diet). This study provides a technical reference for HT and CT to improve the intestinal inflammation of cultured fish caused by terrestrial plant diet, which is of great significance.

The following issues need further discussion.

Point 1: In Table 1, why is the reduction of wheat gluten consumption considered in the design? Does it affect the intestinal inflammation of fish? Suggestions need to be analyzed and explained in the discussion.

Response 1: The maximum variation of the weath gluten content in the considered diets was 0.34%, ranging from 4.86 % to 5.20 %, for TSP and Control diets, respectively. This is a pretty neglectable difference which determine a starch content variation of 0.03% only (Table 1). However, these variations were necessary for compensate the different inclusion of TSP in the experimental diet (the total amount of the ingredients must be equal to 100). To our knowledge, this modification was the one that to a lesser extent modified the nutritional and organoleptic characteristic of the diets. Modifying the inclusion of other functional raw materials there would have been no higher certainty about the entity of possible unwanted (for experimental reasons) effects on the observed parameters. In conclusion, sure, we did not modify the weath gluten content because its possible pro-inflammatory effect.

Point 2: On page 7, I suggest that the organizational forms of A and C in Figure 1 should be the same as those of B and D, so that the comparison will be clearer.

Response 2: Thanks for the suggestion. We have changed B and D images with two new images showing organizational forms more similar to A and C.

Point 3: In "4 Discussion" on page 12, the second paragraph (lines 402-410) is suggested to be introduced in the foreword, which directly discusses the research results.

 Response 3: Thanks for the suggestion. We moved lines 402-410 in the foreword and modify the second papagraph.

Point 4: On page 14, lines 493-506, it is unclear whether the content of Fusobacterium and other bacteria that can cause intestinal inflammation in the intestine of zebrafish fed with tannin diet increases or decreases, and whether they have anti-inflammatory or inflammatory effects.

Response 4: Thank you for highlighting this point which has give us the possibility to improve the discussion section. The sentences reported in the lines 493-506 were revised specifying the pro- or anti-inflammatory role of the main changed bacteria.

Point 5: It is recommended that only the particularly relevant literature be cited, and some are repeated, such as literature [1] and [2]. Only [1] can be cited.

Response 5: We have checked all the literature reported and changed the reported literature where necessary. The new references list is reported too.